# Exploring Temporal Semantic for Incomplete Clustering

## Abstract

Clustering data with incomplete features has garnered considerable scholarly attention; however, the specific challenge of clustering sequential data with missing attributes remains largely under-explored. Conventional heuristic methods generally address this issue by first imputing the missing features, thereby making the clustering results heavily reliant on the quality of imputation. In this paper, we introduce a novel clustering framework, termed *ETC-IC*, which directly clusters incomplete data with rigorous theoretical guarantees, whilst concurrently leveraging temporal semantic consistency to enhance clustering performance. Empirical evaluations demonstrate that the proposed model consistently surpasses current state-of-the-art methods in clustering human motion data.

## 1 Introduction

Subspace clustering serves as a fundamental tool in data analysis, modelling data as arising from a union of lower-dimensional subspaces Xia et al. (2017). Formally, consider a dataset in $\mathbb{R}^D$, comprising $N$ instances, denoted as $\{\mathbf{x}_n \in \mathbb{R}^D\}_{n=1}^N$. These data points are assumed to lie within a union of $K$ subspaces, represented by $\{\mathcal{S}_k\}_{k=1}^K$, each with an unknown dimension $d_k = \dim(\mathcal{S}_k)$, where $0 < d_k < D$. The objective is to learn both the subspace features of unknown dimension and the corresponding clustering assignment.

Despite the recent emergence of various subspace clustering methodologies Wang et al. (2023a); Li et al. (2023); Fettal et al. (2023); Mo & Raj (2024); Tang et al. (2024); Li et al. (2024); Ma et al. (2024); Gong et al. (2024), comparatively little attention has been devoted to the study of clustering with incomplete data. Although a few approaches Dung et al. (2021); Mahmood & Pimentel-Alarcón (2022); Soni et al. (2023) have been proposed to address subspace clustering with missing entries, these methods generally assume that data points are independently sampled from multiple subspaces, thereby neglecting the explicit temporal information inherent in sequential data. For instance, in the clustering of human motion data, once a particular motion begins, it typically persists for a certain duration before transitioning—this temporal continuity, intrinsic to such datasets, is of significant importance. Effectively leveraging this temporal information is crucial for the successful clustering of sequential data. However, capturing this discriminative temporal information remains an arduous challenge, primarily due to the intricacies of temporal dependencies and the complexity involved in addressing missing features while exploring temporal semantics in a principled manner.

This paper introduces the *Exploring Temporal Semantic for Incomplete Clustering (ETC-IC)* framework, which possesses the capability to seamlessly integrate temporal information while concurrently addressing the challenge of missing data. Firstly, to manage the issue of missing entries, we employ an algebraic subspace analysis and develop a theoretically grounded alternative, thereby ensuring accurate clustering even in the presence of incomplete data. Secondly, we explore the temporal semantics inherent in sequential data by aligning data points and their temporal assignments through a temporal semantic consistency constraint, thereby ensuring that data points with similar temporal semantics are clustered together. The handling of missing data and the exploration of temporal semantics are unified within a single cohesive framework, thereby demonstrating the adaptability and versatility of the proposed method in addressing incomplete sequential data as required.

In summary, the principal contributions of this paper are as follows:

- **A Novel Clustering Framework for Incomplete Sequential Data:** We present a clustering framework distinguished by its remarkable adaptability in addressing the inherent challenges posed by incomplete sequential data.

- **Temporal Semantic Consistency in Clustering:** We introduce an innovative temporal semantic consistency constraint, which markedly enhances the efficacy of subspace clustering for sequential data.

- **Theoretical Analysis of Clustering Incomplete Sequential Data:** We provide a rigorous theoretical analysis, enabling an equivalent approach even in the presence of missing data, whilst effectively exploring temporal semantics.

Unsupervised human motion segmentation is fundamental to the automatic discovery and comprehension of complex motion patterns without the need for labelled data Kuehne et al. (2011); Martinez et al. (2017); MacKenzie (2024). Over the past decade, research has demonstrated that subspace clustering yields promising results for this task Xia et al. (2017); Keuper et al. (2018); Zhou et al. (2022). Nevertheless, few studies have addressed clustering in the presence of missing pixels or frames. This paper employs five benchmark datasets for human motion segmentation to evaluate the proposed method, demonstrating that *ETC-IC* consistently outperforms state-of-the-art techniques in scenarios involving incomplete data.

## 2 RELATED WORKS

### 2.1 INCOMPLETE CLUSTERING

Clustering data with missing entries remains a critical domain within machine learning, wherein the challenge lies in effective clustering despite incomplete observations. Enhanced methodologies such as *FGSSC* employ a greedy strategy to mitigate errors by treating them as erasures Petukhov & Kozlov (2015). *SSC-EWZF* extends clustering capabilities by estimating a kernel matrix based upon available observations Yang et al. (2015). *SCMD* and *GSSC-MD* ensure subspace identification through information-theoretic conditions and group-sparse regularisation, respectively Pimentel-Alarcon & Nowak (2016); Pimentel-Alarcón et al. (2016). *PTSC* leverages Gaussian Process priors for effective data segmentation Gholami & Pavlovic (2017), whilst *PZF-SSC* projects data onto observed coordinate subspaces for enhanced clustering efficacy Tsakiris & Vidal (2018). Recent advancements have increasingly focused on robust and versatile approaches, such as Non-Convex Fusion Penalty Clustering (*NCFPC*), which employs $\ell_0$ penalties to induce sparsity Poddar & Jacob (2019), and Deep Structure-Preserving Autoencoders (*DSPA*), which project incomplete data into a latent space whilst preserving its intrinsic geometric structure Choudhury & Pal (2019). *PETRELS*, integrating ADMM with PETRELS, addresses the challenges of outlier detection and missing entries Dung et al. (2021). *FSC* reduces inter-subspace distances to achieve effective clustering despite data gaps Mahmood & Pimentel-Alarcón (2022), while *MISS-DSG* employs a mixed-integer framework to optimise subspace assignment Soni et al. (2023).

However, existing research lacks a clustering framework equipped with the capacity to address, in an optional manner, both incomplete data and sequential data, whilst providing rigorous theoretical guarantees.

### 2.2 CLUSTERING LEVERAGING TEMPORAL INFORMATION

Clustering techniques that harness temporal information have demonstrated considerable efficacy in the context of sequential data clustering. Zhou et al. Zhou et al. (2012) introduced a framework for segmenting time series into meaningful clusters. The Ordered Subspace Clustering (OSC) method Tierney et al. (2014) employs a consistency constraint to ensure temporal coherence, while Temporal Subspace Clustering (TSC) Li et al. (2015) integrates non-negative dictionary learning with temporal Laplacian regularisation. The Low-Rank Transfer Subspace (LTS) approach Wang et al. (2018b) captures temporal correlations via a graph regulariser, whereas Consistency and Diversity Induced Clustering (CDMS) Zhou et al. (2022) utilises transfer subspace learning for video data. Other noteworthy methodologies, including Dual-Side Auto-Encoder (DSAE) Bai et al. (2020), Deep Video Action Clustering (DVAC) Peng et al. (2021), and Velocity-Sensitive Dual-Side Auto-

Encoder (VSDA) Bai et al. (2022), enhance representation learning by integrating spatio-temporal features and temporal consistency strategies.

However, these methods primarily focus on data preprocessing without adequately exploring the influence of temporal semantics on clustering Bai et al. (2022); Zhou et al. (2022); Wang et al. (2023b). Consequently, they often fall short of ensuring that clustering outcomes faithfully capture the temporal subtleties inherent in sequential data. Furthermore, existing works necessitate data imputation prior to processing, thereby rendering their clustering performance highly susceptible to the quality of data imputation.

## 2.3 UNSUPERVISED HUMAN MOTION SEGMENTATION

Unsupervised methodologies for human motion segmentation employ a diverse range of techniques to adeptly process intricate, unlabeled dynamic motion data Gong et al. (2013). Progress in clustering has significantly advanced the field; for instance, Zhou et al. Zhou et al. (2012) proposed an unsupervised hierarchical bottom-up clustering framework that partitions a multidimensional time series into distinct segments. Wang et al. Wang et al. (2022b) refine graphs for clustering by removing extraneous connections, Bai et al. Bai et al. (2022) derive neighbor consistency features, and Zhou et al. Zhou et al. (2022) utilize a multi-mutual consistency learning strategy for decomposing multi-layer feature spaces in affinity matrix construction. Significant contributions to enhancing human motion segmentation include Zhu et al.'s Zhu et al. (2023) adaptive local-component-aware graph convolutional network, Liang et al.'s Liang et al. (2023) locater with a dual-component memory system, and Shi et al.'s Shi et al. (2023) triDet framework, which incorporates a trident-head and scalable-granularity perception layer. Despite these advancements, considerable challenges persist in effectively harnessing temporal semantics to further improve human motion segmentation, particularly in scenarios involving missing entries within human motion data.

# 3 METHODOLOGY

## 3.1 UNION OF SUBSPACES MODEL

Consider utilizing subspaces to approximate the data manifold $\mathbf{x} \in \mathbb{R}^D$. Let $\mathbf{U} \in \mathbb{R}^{D \times d}$ be a basis matrix composed of $d$ columns $\{\mathbf{u}_i\}_{i=1}^d$. Furthermore, let $\mathbf{o} \in \mathbb{R}^D$ denote the offset of the affine subspace, which must be orthogonal to the columns of $\mathbf{U}$ to ensure its uniqueness, i.e., $\mathbf{U}^T \mathbf{o} = \mathbf{0}$. The affine subspace model readily reduces to the linear subspace model when $\mathbf{o} = \mathbf{0}$, thereby allowing the model to effectively capture both linear and non-linear unions of subspaces (UoS).

A data point residing within a subspace can be modeled as $\mathbf{x} = \mathbf{U}\mathbf{v} + \mathbf{o} + \boldsymbol{\epsilon}$, where $\boldsymbol{\epsilon}$ represents a stochastic term accounting for noise or bias due to subspace approximation errors. This noise component $\boldsymbol{\epsilon}$ is assumed to have zero mean and is orthogonal to the columns of $\mathbf{U}$. The affine span is thereby defined as $\mathcal{S}_a(\mathbf{U}, \mathbf{o}) = \{\mathbf{x} \mid \mathbf{x} = \mathbf{U}\mathbf{v} + \mathbf{o} + \boldsymbol{\epsilon}, \mathbf{v} \in \mathbb{R}^d\}$, where $\mathbf{v} \in \mathbb{R}^d$ are the subspace coefficients for $\mathbf{x}$. We assume the data points reside within a union of subspaces $\{\mathcal{S}_a(\mathbf{U}_k, \mathbf{o}_k)\}_{k=1}^K$, where $\mathbf{U}_k = [\mathbf{u}_{k,1}, \ldots, \mathbf{u}_{k,d_k}] \in \mathbb{R}^{D \times d_k}$ represents the basis of the $k$-th subspace and $\mathbf{o}_k \in \mathbb{R}^D$ its corresponding offset.

Partitioning the data into segments based on the learned UoS model begins with expressing $\boldsymbol{\epsilon}$ as $\boldsymbol{\epsilon} = \mathbf{x} - \mathbf{U}\mathbf{v} - \mathbf{o}$. Since $\boldsymbol{\epsilon}$ is orthogonal to $\mathbf{U}$, we have $\mathbf{U}^T(\mathbf{x} - \mathbf{o} - \mathbf{U}\mathbf{v}) = \mathbf{0}$, leading to the solution for $\mathbf{v}$:

$$\mathbf{v} = (\mathbf{U}^T\mathbf{U})^{-1}\mathbf{U}^T(\mathbf{x} - \mathbf{o}).$$

Substituting $\mathbf{v}$ back yields the residual $\boldsymbol{\epsilon}$ as:

$$\boldsymbol{\epsilon} = \mathbf{x} - \mathbf{U}(\mathbf{U}^T\mathbf{U})^{-1}\mathbf{U}^T(\mathbf{x} - \mathbf{o}) - \mathbf{o}.$$

The squared distance between a data point $\mathbf{x}$ and the subspace $\mathcal{S}_a(\mathbf{U}, \mathbf{o})$ is then given by $\|\boldsymbol{\epsilon}\|_2^2$, which defines the subspace residual function:

$$\varepsilon(\mathbf{x}, \mathbf{U}, \mathbf{o}) = \|\boldsymbol{\epsilon}\|_2^2 = \|\mathbf{P}_\perp(\mathbf{x} - \mathbf{o})\|_2^2, \tag{1}$$

where $\mathbf{P}_\perp = \mathbf{I}_D - \mathbf{U}(\mathbf{U}^T\mathbf{U})^{-1}\mathbf{U}^T$ projects onto the orthogonal complement of the subspace spanned by $\mathbf{U}$.

Now, consider the scenario where some entries of the data point $\mathbf{x}$ are missing. Let $\Phi$ be a $D \times D$ diagonal matrix representing the non-missing indicator, where $\phi_j = 1$ if the $j$-th entry is observed and $\phi_j = 0$ otherwise. Define $m = \sum_{j=1}^{D} \phi_j$ as the number of observed entries. The residual for $\mathbf{x}$ in the presence of missing data is given by:

$$\xi(\mathbf{x}, \Phi, \mathbf{U}, \mathbf{o}) = \varepsilon(\Phi\mathbf{x}, \Phi\mathbf{U}, \Phi\mathbf{o}) = \|\mathbf{P}_\perp^\Phi \Phi(\mathbf{x} - \mathbf{o})\|_2^2 \tag{2}$$

where $\mathbf{P}_\perp^\Phi = \mathbf{I}_D - \Phi\mathbf{U}(\mathbf{U}^\mathrm{T}\Phi\mathbf{U})^{-1}\mathbf{U}^\mathrm{T}\Phi$.

**Theorem 1.** *If $\varepsilon(\mathbf{x}, \mathbf{U}, \mathbf{o}) \neq 0$, then there exist constants $\beta_1, \beta_2 \geq 0$ and a finite constant $C$ such that, with probability at least $1 - 4\delta$,*

$$\frac{m}{D} - \beta_1 \leq \frac{\xi(\mathbf{x}, \Phi, \mathbf{U}, \mathbf{o})}{\varepsilon(\mathbf{x}, \mathbf{U}, \mathbf{o})} \leq \frac{m}{D} + \beta_2$$

*holds when $m \geq C$. (Proof provided in Appendix A.1.)*

Theorem 1 asserts that $\varepsilon(\mathbf{x}, \mathbf{U}, \mathbf{o})$ approximates $\frac{D}{m}\xi(\mathbf{x}, \Phi, \mathbf{U}, \mathbf{o})$ with high probability when $m$ exceeds a certain threshold. Moreover, if $\varepsilon(\mathbf{x}, \mathbf{U}, \mathbf{o}) = 0$, then it follows that $\xi(\mathbf{x}, \Phi, \mathbf{U}, \mathbf{o}) = 0$ as well.

## 3.2 Temporal Semantics Consistency

Our objective is to assign the data points $\{\mathbf{x}_1, \ldots, \mathbf{x}_N\}$, which may contain missing entries, to subspaces $\{\mathcal{S}_a(\mathbf{U}_k, \mathbf{o}_k)\}_{k=1}^{K}$ while ensuring that each data point and its neighbors with the same temporal semantic belong to the same subspace. However, the temporal boundaries and durations of clustering semantics are unknown, which poses a challenge to achieving temporal semantic consistency on clustering assignment.

It is observed that the data point at time $t$ is typically assigned to the same cluster as its preceding and succeeding points. By considering these neighboring data points during subspace assignment, the accuracy of clustering can be substantially improved. Thus, we propose a temporal semantics consistency constraint, which enforces the assignment of a data point and its temporal neighbors to the same subspace.

We propose an automatic discriminative searching scheme to determine the neighbors of each data point. Mathematically, each $\mathbf{x}_i$ is encouraged to be clustered together with its nearest sequential neighbors. The right bound for the $i$th sample $r_i$ is equal to the min $j \in \{i+1, i+2, ..., N\}$ that satisfies $\|\mathbf{x}_j - \mathbf{x}_{j+1}\|_2 > \|\mathbf{x}_{j-1} - \mathbf{x}_j\|_2$ and $\|\mathbf{x}_j - \mathbf{x}_{j+1}\|_2 > \|\mathbf{x}_{j+1} - \mathbf{x}_{j+2}\|_2$. The left bound for the $i$th sample $l_i$ is equal to the max $j \in \{1, 2, ..., i-1\}$ that satisfies $\|\mathbf{x}_{j-1} - \mathbf{x}_j\|_2 > \|\mathbf{x}_{j-2} - \mathbf{x}_{j-1}\|_2$ and $\|\mathbf{x}_{j-1} - \mathbf{x}_j\|_2 > \|\mathbf{x}_j - \mathbf{x}_{j+1}\|_2$. The physical implication is that a data point $\mathbf{x}$ is close to its temporal neighbors. The neighbors of $\mathbf{x}$'s neighbors are still considered $\mathbf{x}$'s neighbors, and so on, until no further neighbors can be identified.

Suppose $\mathcal{N}_i$ saves the index of spatio-temporal neighbors of the $i$th data point, and $\mathcal{N}_i = \{j | j \in \{l_i, l_i + 1, ..., r_i\}, j \neq i\}$. It is noteworthy that the neighbor set $\mathcal{N}_i$ is determined automatically, requiring no manual parameter tuning. Consider using the data point whose index is in $\mathcal{N}_i$ to guide the assignment of the data point $\mathbf{x}_i$. If $\mathbf{x}_i$ is located in subspace $\mathcal{S}(\mathbf{U}_k, \mathbf{o}_k)$, then its neighbors in $\mathcal{N}_i$ are also encouraged to be located in subspace $\mathcal{S}(\mathbf{U}_k, \mathbf{o}_k)$. The neighborhood cost for the data point $\mathbf{x}_i$ is the sum of distances between neighbors in $\mathcal{N}_i$ and the subspace $\mathcal{S}(\mathbf{U}_k, \mathbf{o}_k)$, i.e., $\sum_{j \in N_i} \frac{D}{m_j} \xi(\mathbf{x}_j, \Phi_j, \mathbf{U}_k, \mathbf{o}_k)$.

## 3.3 Learning UoS by Exploiting Temporal Semantics with Missing Entries

We propose the following optimization problem to learn the UoS model while accounting for incomplete data and temporal semantics:

$$\underset{\{\mathcal{C}_k, \mathbf{U}_k, \mathbf{o}_k\}_{k=1}^{K}}{\text{minimize}} \quad \sum_{k=1}^{K} \sum_{i \in \mathcal{C}_k} d_{k,i}, \qquad \text{subject to} \quad \mathbf{U}_k^\mathrm{T}\mathbf{o}_k = \mathbf{0}, \quad \forall k, \tag{3}$$

where

$$d_{k,i} = \frac{D}{m_i}\xi(\mathbf{x}_i, \Phi_i, \mathbf{U}_k, \mathbf{o}_k) + \frac{1}{|\mathcal{N}_i|}\sum_{j \in \mathcal{N}_i} \frac{D}{m_j}\xi(\mathbf{x}_j, \Phi_j, \mathbf{U}_k, \mathbf{o}_k),$$

with $|\mathcal{N}_i|^{-1}$ serving to balance the influence of the temporal semantic consistency constraint.

The objective function of problem (3) promotes data points being proximate to their assigned subspaces while also enforcing their temporal neighbors to reside within the same subspace. Although problem (3) is inherently non-convex, we employ a transformation, as detailed in the following theorem, to render the optimization tractable.

**Theorem 2.** *The term $d_{k,i}$ in equation (3) can be equivalently expressed as*

$$\sum_{i=1}^{N} \frac{D}{m_i} \xi(\mathbf{x}_i, \Phi_i, \mathbf{U}_k, \mathbf{o}_k)\big(\mathbb{I}(i \in \mathcal{C}_k) + n_k(i)\big),$$

*where $n_k(i)$ represents the number of occurrences of the data point $\mathbf{x}_i$ as a spatio-temporal neighbor of another data point within the $k$-th subspace. Specifically, $n_k(i) = \sum_{j \in \mathcal{C}_k} \frac{1}{|\mathcal{N}_j|}\mathbb{I}(i \in \mathcal{N}_j)$, and $\mathbb{I}(s)$ is an indicator function which evaluates to $1$ if the condition $s$ holds, and $0$ otherwise. (The proof is provided in the Appendix A.2.)*

Based on Theorem 2, the optimization problem (3) can be reformulated as follows:

$$\underset{\{\mathcal{C}_k, \mathbf{U}_k, \mathbf{o}_k\}_{k=1}^{K}}{\text{minimize}} \quad \sum_{k=1}^{K} \sum_{i=1}^{N} \frac{D}{m_i} \xi(\mathbf{x}_i, \Phi_i, \mathbf{U}_k, \mathbf{o}_k)w_{k,i}, \quad \text{subject to} \quad \mathbf{U}_k^{\mathsf{T}}\mathbf{o}_k = \mathbf{0}, \quad \forall k, \quad (4)$$

where $w_{k,i} = \mathbb{I}(i \in \mathcal{C}_k) + n_k(i)$.

### 3.4 AN ALTERNATING OPTIMIZATION ALGORITHM

Observe that problem (4) contains two variable blocks. The variables $\{\mathbf{U}_k, \mathbf{o}_k\}_{k=1}^{K}$ depend upon the subspace assignment $\{\mathcal{C}_k\}_{k=1}^{K}$, and conversely, $\{\mathcal{C}_k\}_{k=1}^{K}$ is contingent upon $\{\mathbf{U}_k, \mathbf{o}_k\}_{k=1}^{K}$. Hence, an alternating optimization approach is well-suited for solving this problem. Initially, we solve for $\{\mathbf{U}_k, \mathbf{o}_k\}$ given the subspace assignment $\{\mathcal{C}_k\}_{k=1}^{K}$. Let the objective function in problem (4) be denoted as $\mathcal{J}(\{\mathbf{U}_k, \mathbf{o}_k\}_{k=1}^{K})$. By differentiating $\mathcal{J}(\{\mathbf{U}_k, \mathbf{o}_k\}_{k=1}^{K})$ with respect to $\mathbf{o}_k$ and equating it to zero, we obtain the solution $\mathbf{o}_k'$, whose $j$-th element is given by:

$$o_{k,j}' = \frac{\sum_{i=1}^{N} \frac{w_{k,i}}{m_i} \phi_{i,j} x_{i,j}}{\sum_{i=1}^{N} \frac{w_{k,i}}{m_i} \phi_{i,j}}$$

Note that while $\mathbf{o}_k'$ might not satisfy the original constraint of problem (4), it must still lie within the subspace $\mathcal{S}_a(\mathbf{U}_k, \mathbf{o}_k)$.

---

**Algorithm 1** ETC-IC algorithm.

---

1: **Input:** $\mathbf{X} \in \mathbb{R}^{D \times N}$.
2: Generate $K$ orthogonal subspaces randomly.
3: Initialize $w_{j,i} = 1$ for any $j, i$ and $\mathcal{N}_i$.
4: **repeat**
5:     **for** $k = 1$ **to** $K$ **do**
6:         $\mathcal{C}_k \leftarrow \{i \in \{1, 2, ..., N\} : k = l_i\}$
7:     **end for**
8:     **for** $k = 1$ **to** $K$ **do**
9:         1) Diagonalize $\mathbf{W}_k$ with $w_{k,i}$.
10:        2) Calculate $\mathbf{o}_k'$.
11:        3) Construct subspace base $\mathbf{U_k}$.
12:        4) Calculate $\mathbf{o}_k$.
13:     **end for**
14: **until** the objective function in (4) cannot be decreased.
15: **Output:** $\{\mathcal{C}_k\}_{k=1}^{K}$.

---

Define the mean-shifted data matrix for the $k$-th subspace as $\bar{\mathbf{X}}_k = [\bar{\mathbf{x}}_{k,1}, \bar{\mathbf{x}}_{k,2}, \ldots, \bar{\mathbf{x}}_{k,N}] \in \mathbb{R}^{D \times N}$, where $\bar{\mathbf{x}}_{k,i} = \frac{\mathbf{x}_i - \mathbf{o}_k'}{\sqrt{m_i}} = [\bar{x}_{k,i}^{(1)}, \bar{x}_{k,i}^{(2)}, \ldots, \bar{x}_{k,i}^{(D)}]$. Let $\mathbf{W}_k$ be an $N \times N$ diagonal matrix whose diagonal elements are $w_{k,1}, w_{k,2}, \ldots, w_{k,N}$. Consequently, problem (4) can be reformulated as:

$$\underset{\mathbf{U}_k \in \mathbb{R}^{D \times d_k}}{\arg\min} \sum_{i=1}^{N} \|(\mathbf{I}_D - \Phi_i \mathbf{U}_k (\mathbf{U}_k^{\mathsf{T}} \Phi_i \mathbf{U}_k)^{-1} \mathbf{U}_k^{\mathsf{T}})\Phi_i \bar{\mathbf{x}}_{k,i} w_{k,i}^{1/2}\|_2^2 \quad (5)$$

The optimization problem (5) is equivalent to maximizing $\text{Tr}(\mathbf{U}_k^{\mathsf{T}} \mathbf{S}_k \mathbf{U}_k)$, where the $(a,b)$-th element of $\mathbf{S}_k$ is defined as:

$$(\mathbf{S}_k)_{a,b} = \frac{\sum_{i=1}^{N} w_{k,i} \phi_{i,a} \phi_{i,b} \bar{x}_{k,i}^{(a)} \bar{x}_{k,i}^{(b)}}{\sum_{i=1}^{N} w_{k,i} \phi_{i,a} \phi_{i,b}}$$

The columns of $\mathbf{U}_k$ are required to be orthonormal, i.e., $\mathbf{U}_k^T\mathbf{U}_k = \mathbf{I}_{d_k}$. The solution to (5) is found by selecting the eigenvectors of $\mathbf{S}_k$ corresponding to the $d_k$ largest eigenvalues. It is worth noting that if $\Phi_i = \mathbf{I}_D$, then $\mathbf{S}_k = \bar{\mathbf{X}}_k\mathbf{W}_k\bar{\mathbf{X}}_k^T$. The proof follows standard principles from statistical signal processing theory, such as those found in Kay (1993).

A pertinent question in equation (5) is how to determine $\hat{d}_k$ for all $k = 1, 2, \ldots, K$. In practical applications, assuming the subspace dimensions $d_1, d_2, \ldots, d_K$ are known in advance is often unrealistic Vidal (2011). Even if these dimensions are known but differ, the challenge of establishing a one-to-one correspondence between $\{d_k\}_{k=1}^K$ and $\{\hat{d}_k\}_{k=1}^K$ persists, given that the cluster permutations are unknown. To address this, we propose an adaptive strategy to select $\hat{d}_k$ for all $k = 1, 2, \ldots, K$. Specifically, let $\lambda_{k,1} \geq \lambda_{k,2} \geq \cdots \geq \lambda_{k,D}$ represent the $D$ leading eigenvalues of $\mathbf{S}_k$. Then, for all $k = 1, 2, \ldots, K$, we set:

$$\hat{d}_k = \arg\max_{i \in \{1,2,\ldots,D-1\}} (\lambda_{k,i} - \lambda_{k,i+1})$$

Since the subspace offset $\mathbf{o}_k'$ may not satisfy the constraint in problem (4), we present the following proposition.

**Proposition 3.** *If $\mathbf{o}_k'$ lies on the affine subspace $\mathcal{S}_a(\mathbf{U}_k, \mathbf{o}_k)$, then $\mathbf{o}_k$ can be expressed as:*

$$\mathbf{o}_k = (\mathbf{I}_D - \mathbf{U}_k(\mathbf{U}_k^T\mathbf{U}_k)^{-1}\mathbf{U}_k^T)\mathbf{o}_k'$$

*satisfying $\mathbf{U}_k^T\mathbf{o}_k = \mathbf{0}$. (Proof is provided in the Appendix A.3.)*

Thus, given $\mathbf{U}_k$ and $\mathbf{o}_k'$, the subspace offset $\mathbf{o}_k$ can be easily computed accordingly.

Next, we address the subspace assignment problem given the subspace information, i.e., fixing $\{\mathbf{U}_k, \mathbf{o}_k\}_{k=1}^K$ and updating $\{\mathcal{C}_k\}_{k=1}^K$ by solving problem (4). This is accomplished by evaluating the weighted combination of the residual from the data point to a given subspace, alongside the residuals of its sequential neighbors, thereby assigning the estimated cluster label of the data point $\mathbf{x}_i$ to the $l_i$-th subspace, where

$$l_i = \arg\min_{k \in \{1,2,\ldots,K\}} \xi(\mathbf{x}_i, \Phi_i, \mathbf{U}_k, \mathbf{o}_k) w_{k,i}.$$

**Theorem 4.** *Given a data point $\mathbf{x}$ and $K$ affine subspaces, let $l = \arg\min_{k \in \{1,2,\ldots,K\}} \varepsilon(\mathbf{x}, \mathbf{U}_k, \mathbf{o}_k)$. If $\varepsilon(\mathbf{x}, \mathbf{U}_l, \mathbf{o}_l) < C_k \varepsilon(\mathbf{x}, \mathbf{U}_k, \mathbf{o}_k)$ for all $k \neq l$, $k \in \{1, 2, \ldots, K\}$, where $C_k \in (0,1)$ is a finite constant, then there exists a finite constant $C_0$ such that:*

$$l = \arg\min_{k \in \{1,2,\ldots,K\}} \xi(\mathbf{x}, \Phi_i, \mathbf{U}_k, \mathbf{o}_k)$$

*holds with probability at least $1 - 4(K-1)\delta$ when $m > C_0$. (Proof is provided in the Appendix A.4.)*

Theorem 4 indicates that if the data point $\mathbf{x}$ is closest to the $l$-th subspace and distant from others, then with high probability, the data point $\mathbf{x}$ with $m$ non-missing entries will also be nearest to the $l$-th subspace when $m$ is sufficiently large. This implies that under minimal data loss, the proposed subspace assignment method remains robust despite missing data.

The pseudo-code for the proposed subspace clustering method is presented in Algorithm 1. Initially, $K$ subspaces are randomly constructed, and then iterative optimization of the clustering assignment is performed until the objective function in problem (4) cannot be further minimized.

## 4 EXPERIMENTAL RESULTS

**Dataset.** We assess the efficacy of the proposed method using five benchmark datasets for human motion segmentation, which have been widely utilized in prior studies Tierney et al. (2014); Li et al. (2015); Wang et al. (2018b); Peng et al. (2021); Wang et al. (2022b; 2018a; 2022a); Cui et al. (2021). The Weizmann (Weiz) dataset Gorelick et al. (2007) comprises 90 human motion sequences, encompassing ten distinct actions such as running, walking, and skipping, executed by nine

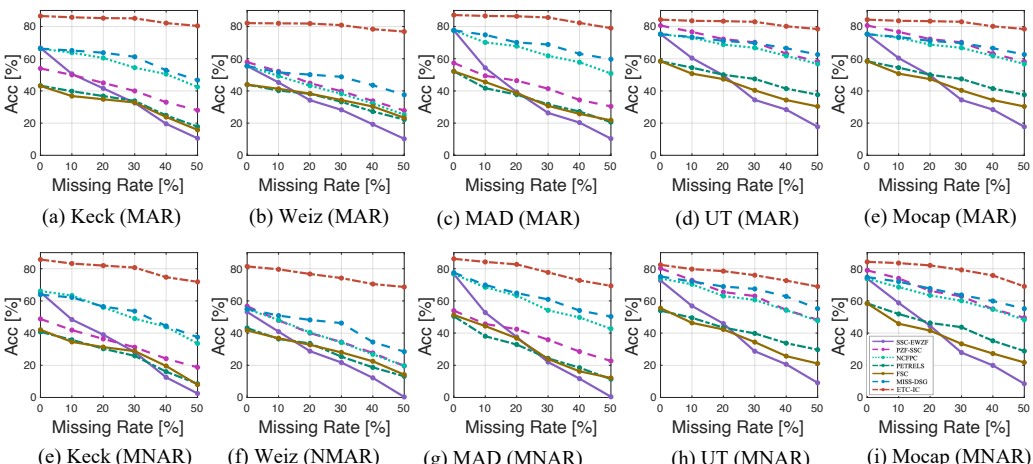

Figure 1: The segmentation performance of different methods on four datasets when the data suffer from pixel-level missing.

subjects in outdoor settings. The Multi-Modal Action Detection (MAD) dataset Huang et al. (2014) includes 35 multi-modal motion sequences from 20 subjects, recorded in three different formats using Microsoft Kinect. The Keck Gesture (Keck) dataset Jiang et al. (2012) consists of 14 distinct motions derived from military signal gestures, performed by three individuals. The UT-Interaction (UT) dataset Ryoo & Aggarwal (2009) features 20 human motion sequences, each illustrating one of six categories of human interaction, including punching, kicking, pushing, hugging, pointing, and handshaking. Lastly, the Carnegie Mellon Action Capture (Mocap) dataset comprises skeletal measurements from 149 subjects engaged in a diverse array of activities, with data selected from five individuals performing between five to twelve actions, thereby providing comprehensive positional and joint angle measurements over various temporal instances.

To demonstrate the effectiveness of the proposed method in managing partially observed data, we conducted experiments under two distinct *pixel-level* scenarios of missing entries: Missing at Random (MAR) and Missing Not at Random (MNAR). In the MAR scenario, pixels were randomly omitted from each frame until the missing rate reached a predefined threshold. For the MNAR scenario, we systematically removed multiple $20 \times 20$ pixel blocks from various regions within each frame, repeating this process until the cumulative proportion of missing pixels reached the designated threshold. Furthermore, we examined the *frame-level* scenario, wherein entire frames were missing. Under the MAR condition, frames were randomly removed until the specified missing rate was attained, whereas in the MNAR condition, consecutive sequences of ten frames were removed at random, ensuring that the overall frame missing rate was not less than the predetermined threshold.

**Compared Methods.** We evaluated the proposed method against clustering approaches capable of handling missing entries, including SC-EWZF Yang et al. (2015), PZF-SSC Tsakiris & Vidal (2018), NCFPC Poddar & Jacob (2019), PETRELS Dung et al. (2021), FSC Mahmood & Pimentel-Alarcón (2022), and MISS-DSG Soni et al. (2023). For a comprehensive description of these methods, please refer to Sec. 2.1. In all experiments, clustering accuracy (Acc) was employed as the primary evaluation metric.

### 4.1 PERFORMANCE OF SUBSPACE CLUSTERING WITH MISSING ENTRIES

Fig. 1 illustrates the clustering outcomes (averaged over five trials) for all methods under both pixel-level MAR and MNAR settings across all datasets. The proposed method consistently surpasses the baseline approaches, achieving over 80% accuracy in scenarios without missing data. This superior performance is attributable not only to the inherent robustness of the clustering algorithm but also to the incorporation of the temporal semantic consistency constraint, which shall be further elucidated in the subsequent ablation study. Furthermore, despite a higher propensity for errors in the MNAR scenario compared to the MAR setting, the proposed method demonstrates significantly enhanced

resilience—particularly at elevated missing rates—when compared with all baseline methods. Such robustness is primarily due to the effective strategy for managing missing data, underpinned by the theoretical assurances of our approach.

Fig. 2 (a) and (b) depict the clustering results (averaged over five iterations) for all methods under frame-level MAR and MNAR conditions on the Keck dataset. Initially, linear interpolation was employed to reconstruct the missing frames, following which the proposed method was applied to the completed human motion data. The proposed approach exhibits markedly enhanced robustness—particularly under conditions of higher frame loss—compared to all competing methods across varying levels of missing data. Moreover, all methods demonstrate inferior performance under the MNAR setting in contrast to the MAR setting, principally due to the sequential loss of frames, which disrupts the temporal continuity, thereby impairing the accuracy of human motion segmentation. Notwithstanding this, the proposed ETC-IC consistently attains the highest performance.

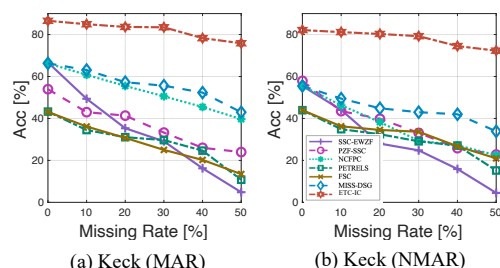

Figure 2: Acc performance on the Keck dataset when the data suffer from frame-level missing.

To further assess the efficacy of the proposed model in addressing human motion data with missing entries, we consider the joint missing scenario, wherein random markers—each representing the three-dimensional spatial coordinates of a human body joint—are absent. In the Mocap dataset, each frame typically consists of 31 to 41 joints, representing various segments of the human anatomy. The data for these joints encompass positional and angular information in three-dimensional space, with the precise count depending on the specific motion and recording configuration. The yellow skeleton in Fig. 3(a) illustrates the missing joints resulting from marker absence. Fig. 3(b) displays the performance of the proposed method in comparison with baseline approaches across different levels of missing markers. The proposed ETC-IC method consistently exhibits the most stable accuracy, irrespective of the degree of missing data.

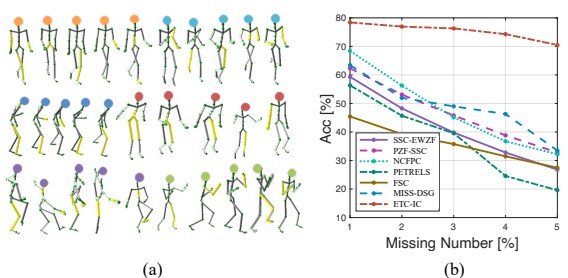

Figure 3: (a) Skeleton action with 1∼3 missing joints; (b) Performance with skeleton missing on Mocap dataset.

### 4.2 QUANTITATIVE RESULT

In Fig. 4, we illustrate the human motion segmentation outcomes obtained by the proposed ETC-IC method in comparison with several baseline approaches on the Keck dataset. The

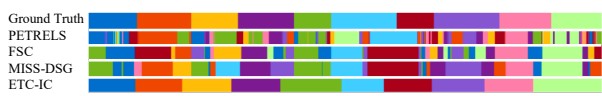

Figure 4: Visualization of clustering results on Keck dataset. The 10 colors denote 10 different action clusters.

baselines, which include PETRELS Dung et al. (2021), FSC Mahmood & Pimentel-Alarcón (2022), and MISS-DSG Soni et al. (2023), treat each sample independently, leading to disordered clustering results that inadequately reflect temporal continuity. Such methods frequently encounter difficulties in preserving the integrity of individual clusters, often fragmenting them into multiple segments. In stark contrast, our approach yields segments that are both continuous and coherent, thereby producing clustering outcomes characterized by distinctly preserved temporal semantics.

To further elucidate the efficacy of the proposed models, we commence by presenting visualizations of representative frames from the human motion data in the Weiz dataset, as illustrated in Fig. 5. Fig. 6(a) portrays the convergence of the proposed method on the objective function value, evidencing that the proposed method converges swiftly within 10 iterations. Thereafter, we depict the human motion segmentation results at each iteration, as demonstrated in Fig. 6(b). The proposed approach attains convergence after 10 iterations, with substantial adjustments taking place during the initial three iterations. In subsequent iterations, the segmentation stabilities, followed by a gradual refinement of boundaries. Upon completion of these boundary adjustments, the algorithm reaches convergence.

In comparison with the ground truth, segmentation errors predominantly occur at the transition boundaries between distinct motions, such as between the seventh and eighth actions, specifically 'jumping jack' and 'jump'. We have discerned that the principal factors hindering clustering efficacy are the stability of the frame background and the explicitness of the actions. Fig. 7 presents visual examples of misclassified segments. As the 'jumping jack' action contains frames that are visually similar to the initial frames of the 'jump' action, the proposed method occasionally misclassified the commencement of the 'jump' action as part of the 'jumping jack' action. The principal challenge in clustering actions lies in the inherently ambiguous nature of some actions, which often results in ill-defined boundaries between them. While the clustering results may deviate from the ground truth, such deviations do not inherently indicate that the results are unreasonable.

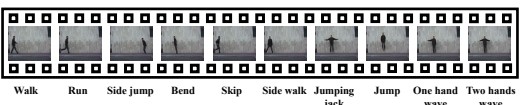

Figure 5: Visualizations of example frames depicting different motions of Person 'ido' in the Weiz Dataset.

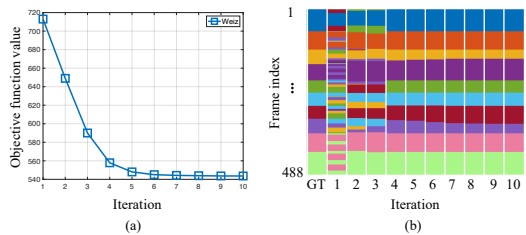

Figure 6: (a) Convergence demonstration of the proposed method on the human motion data in the Weiz dataset. (b) Visualizations of the dynamic clustering assignment to subspaces during optimization on the Weiz dataset. Different colors represent different clusters/motions. GT stands for 'ground truth'.

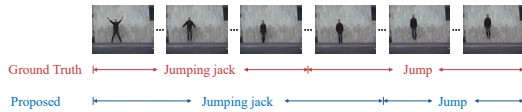

Figure 7: Visualizations of false cases in human motion segmentation on the human motion sequence in Weiz dataset.

### 4.3 ABLATION STUDY

Tab. 8 presents an ablation study on the proposed *ETC-IC* method, assessing the significance of incorporating temporal semantics. We evaluated the impact of the temporal semantic consistency constraint on the efficacy of *ETC-IC* across four datasets, each containing 10% missing at random (MAR) data. The results indicate that the removal of this constraint results in a marked decline in the performance of the proposed method. Nevertheless, even without temporal exploration, *ETC-IC* continues to outperform the state-of-the-art techniques.

Figure 8: Ablation study of the proposed ETC-IC on temporal semantics.

| | Keck | Weiz | MAD | UT | Mocap |
|---|---|---|---|---|---|
| Non-temporal semantics | 76.2 | 75.4 | 71.4 | 69.7 | 76.1 |
| Proposed | 86.3 | 81.1 | 85.1 | 83.5 | 82.9 |

### 5 CONCLUSION

This paper introduces a clustering method that possesses the capability to seamlessly integrate temporal information whilst concurrently addressing the challenge of missing data. We begin by aligning data points with their temporal dependencies through the imposition of a temporal semantic consistency constraint, followed by an algebraic subspace analysis. A theoretically rigorous solution algorithm is then developed for an equivalent form, ensuring precise clustering results even in the presence of incomplete data. Comprehensive experiments conducted on five benchmark human action datasets consistently demonstrate the superiority of the proposed method.

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

## A APPENDIX

### A.1 PROOF OF THEOREM 1

For the sake of written convenience, we use $\mathcal{S}$ to represent $\mathcal{S}_a(\mathbf{U}, \mathbf{o})$ in the appendix section. In order to detect from a very small number of frames whether there is energy in a vector $\mathbf{x}$ outside the $d$-dimensional subspace $\mathcal{S}$, we must first quantify how much information we can expect each frame to provide. The authors in Candes & Recht (2012) defined the coherence of a subspace $\mathcal{S}$ to be the quantity

$$\rho(\mathcal{S}) := \frac{D}{d} \max_{j \in \{1,2,\dots,D\}} ||\mathbf{U}(\mathbf{U}^{\mathrm{T}}\mathbf{U})^{-1}\mathbf{U}^{\mathrm{T}}\mathbf{e}_j||_2^2 \tag{6}$$

where $\mathbf{e}_j \in \mathbb{R}^D$ is all zero vector expect the $j$th element is one. That is, $\rho(\mathcal{S})$ measures the maximum magnitude attainable by projecting a standard basis element onto $\mathcal{S}$. Note that $1 \leq \rho(\mathcal{S}) \leq D$. For a vector $\boldsymbol{\tau}$, we let $\rho(\boldsymbol{\tau})$ denote the coherence of the subspace spanned by $\boldsymbol{\tau}$. By plugging in the definition, we have

$$\rho(\boldsymbol{\tau}) = \frac{D||\boldsymbol{\tau}||_\infty^2}{||\boldsymbol{\tau}||_2^2}$$

Consider $\mathbf{x} = \mathbf{U}(\mathbf{U}^\mathrm{T}\mathbf{U})^{-1}\mathbf{U}^\mathrm{T}(\mathbf{x} - \mathbf{o}) + \mathbf{o} + \boldsymbol{\epsilon}$, where $\mathbf{U}(\mathbf{U}^\mathrm{T}\mathbf{U})^{-1}\mathbf{U}^\mathrm{T}(\mathbf{x} - \mathbf{o}) \in \mathcal{S}$ is the projection of $\mathbf{x} - \mathbf{o}$ on $\mathcal{S}$, and $\boldsymbol{\epsilon} \in \mathcal{S}^\perp$ is the residual vector of $\mathbf{x}$ to $\mathcal{S}$. Since $\mathbf{U}(\mathbf{U}^\mathrm{T}\mathbf{U})^{-1}\mathbf{U}^\mathrm{T}\boldsymbol{\epsilon} = 0$, we have

$$\mathbf{x} - \mathbf{o} - \mathbf{U}(\mathbf{U}^\mathrm{T}\mathbf{U})^{-1}\mathbf{U}^\mathrm{T}(\mathbf{x} - \mathbf{o}) = \boldsymbol{\epsilon} = \boldsymbol{\epsilon} - \mathbf{U}(\mathbf{U}^\mathrm{T}\mathbf{U})^{-1}\mathbf{U}^\mathrm{T}\boldsymbol{\epsilon} \tag{7}$$

For expression convenience, we denote $\mathbf{x}_\Phi = \Phi\mathbf{x}$, $\mathbf{o}_\Phi = \Phi\mathbf{o}$, and $\mathbf{U}_\Phi = \Phi\mathbf{U}$. Thus, for the missing entries case, equation (7) can be expressed as

$$\mathbf{x}_\Phi - \mathbf{o}_\Phi - \mathbf{U}_\Phi(\mathbf{U}_\Phi^\mathrm{T}\mathbf{U}_\Phi)^{-1}\mathbf{U}_\Phi^\mathrm{T}(\mathbf{x}_\Phi - \mathbf{o}_\Phi)$$
$$= \boldsymbol{\epsilon}_\Phi - \mathbf{U}_\Phi(\mathbf{U}_\Phi^\mathrm{T}\mathbf{U}_\Phi)^{-1}\mathbf{U}_\Phi^\mathrm{T}\boldsymbol{\epsilon}_\Phi \tag{8}$$

Note that $\mathbf{U}_\Phi(\mathbf{U}_\Phi^\mathrm{T}\mathbf{U}_\Phi)^{-1}\mathbf{U}_\Phi^\mathrm{T}\boldsymbol{\epsilon}_\Phi \neq 0$.

$$\varepsilon(\boldsymbol{\epsilon}_\Phi, \mathbf{U}_\Phi, \mathbf{o}_\Phi) = ||\boldsymbol{\epsilon}_\Phi||_2^2 - ||\mathbf{L}_\Phi\mathbf{U}_\Phi^\mathrm{T}\boldsymbol{\epsilon}_\Phi||_2^2 \tag{9}$$

where $\mathbf{L}_\Phi^\mathrm{T}\mathbf{L}_\Phi = (\mathbf{U}_\Phi^\mathrm{T}\mathbf{U}_\Phi)^{-1}$. We also have

$$||\mathbf{L}_\Phi\mathbf{U}_\Phi^\mathrm{T}\boldsymbol{\epsilon}_\Phi||_2^2 \leq ||(\mathbf{U}_\Phi^\mathrm{T}\mathbf{U}_\Phi)^{-1}||_2||\mathbf{U}_\Phi^\mathrm{T}\boldsymbol{\epsilon}_\Phi||_2^2 \tag{10}$$

**Lemma 5.** $0 \leq ||(\mathbf{U}_\Phi^T\mathbf{U}_\Phi)^{-1}||_2 \leq \frac{D}{m(1-\gamma)}$ *with probability at least* $1 - \delta$, *provided that* $\gamma < 1$, *where*

$$\gamma = \sqrt{\frac{8d\rho(\mathcal{S})}{3m}\log\left(\frac{2d}{\delta}\right)}. \tag{11}$$

*Proof.* Proof see AppendixB. $\qquad\square$

**Lemma 6.** $0 \leq ||\mathbf{U}_\Phi^T\boldsymbol{\epsilon}_\Phi||_2^2 \leq \frac{(1+\eta)^2 m}{D}\frac{d\rho(\mathcal{S})}{D}||\boldsymbol{\epsilon}||_2^2$ *with probability at least* $1 - \delta$, *where*

$$\eta = \sqrt{2\rho(\boldsymbol{\tau})\log\left(\frac{1}{\delta}\right)}, \quad \rho(\boldsymbol{\tau}) = \frac{D||\boldsymbol{\tau}||_\infty^2}{||\boldsymbol{\tau}||_2^2} \tag{12}$$

*and* $\boldsymbol{\tau} = (\mathbf{I} - \mathbf{U}(\mathbf{U}^T\mathbf{U})^{-1}\mathbf{U}^T)(\mathbf{x} - \mathbf{o})$.

*Proof.* Proof see AppendixC. $\qquad\square$

Since Lemma 5 and Lemma 6, for equation 10, we have $||(\mathbf{U}_\Phi^\mathrm{T}\mathbf{U}_\Phi)^{-1}||_2||\mathbf{U}_\Phi^\mathrm{T}\boldsymbol{\epsilon}_\Phi||_2^2 \leq \frac{D}{m(1-\gamma)}\frac{(1+\eta)^2 m}{D}\frac{d\rho(\mathcal{S})}{D}||\boldsymbol{\epsilon}||_2^2$ with probability at least $1 - 2\delta$. Thus, we have

$$0 \leq ||\mathbf{L}_\Phi\mathbf{U}_\Phi^\mathrm{T}\boldsymbol{\epsilon}_\Phi||_2^2 \leq \frac{D}{m(1-\gamma)}\frac{(1+\eta)^2 m}{D}\frac{d\rho(\mathcal{S})}{D}||\boldsymbol{\epsilon}||_2^2 \tag{13}$$

with probability at least $1 - 2\delta$.

**Lemma 7.** $\frac{m(1-\alpha)}{D}||\boldsymbol{\epsilon}||_2^2 \leq ||\boldsymbol{\epsilon}_\Phi||_2^2 \leq \frac{m(1+\alpha)}{D}||\boldsymbol{\epsilon}||_2^2$ *with probability at least* $1 - 2\delta$, *where*

$$\alpha = \sqrt{\frac{2\rho^2(\boldsymbol{\tau})}{m}\log\left(\frac{1}{\delta}\right)}. \tag{14}$$

*Proof.* Proof see AppendixD. $\qquad\square$

Since Lemma 7 and equation 13, the term $\varepsilon(\boldsymbol{\epsilon}_\Phi, \mathbf{U}_\Phi, \mathbf{o}_\Phi)$ in (9) is bounded by $(\frac{m(1-\alpha)}{D} - \frac{d\rho(\mathcal{S})\frac{(1+\eta)^2}{1-\gamma}}{D})||\boldsymbol{\epsilon}||_2^2 \leq \varepsilon(\boldsymbol{\epsilon}_\Phi, \mathbf{U}_\Phi, \mathbf{o}_\Phi) \leq \frac{m(1+\alpha)}{D}||\boldsymbol{\epsilon}||_2^2$ with probability at least $1 - 4\delta$. Since $||\boldsymbol{\epsilon}||_2^2 = \varepsilon(\mathbf{x}, \mathbf{U}, \mathbf{o})$, we have $(\frac{m(1-\alpha)}{D} - \frac{d\rho(\mathcal{S})\frac{(1+\eta)^2}{1-\gamma}}{D})\varepsilon(\mathbf{x}, \mathbf{U}, \mathbf{o}) \leq \varepsilon(\mathbf{x}_\Phi, \mathbf{U}_\Phi, \mathbf{o}_\Phi) \leq \frac{m(1+\alpha)}{D}\varepsilon(\mathbf{x}, \mathbf{U}, \mathbf{o})$ with probability at least $1 - 4\delta$, where $\alpha, \eta, \gamma, \rho(\mathcal{S})$ are defined in (14), (12), (11), and (6). We also require $m \geq \frac{8}{3}d\rho(\mathcal{S})\log{(\frac{2d}{\delta})}$ to satisfy $\gamma < 1$ in Lemma 5. Thus, we have the following conclusion:

If $\varepsilon(\mathbf{x}, \mathbf{U}, \mathbf{o}) \neq 0$, then there exist small $\beta_1, \beta_2$ and a finite constant $C$ satisfying $\frac{m}{D} - \beta_1 \leq \frac{\xi(\mathbf{x},\Phi,\mathbf{U},\mathbf{o})}{\varepsilon(\mathbf{x},\mathbf{U},\mathbf{o})} \leq \frac{m}{D} + \beta_2$ with $m \geq C$ with probability at least $1 - 4\delta$, where

$$\beta_1 = \frac{m\alpha}{D} + \frac{d\rho(\mathcal{S})\frac{(1+\eta)^2}{1-\gamma}}{D}, \beta_2 = \frac{m\alpha}{D}, C = \frac{8}{3}d\rho(\mathcal{S})\log{(\frac{2d}{\delta})}.$$

## A.2 Proof of Theorem 2

Recall equation 3, the cost of the frame located in the $k$-th subspace $(\mathbf{U}_k, \mathbf{o}_k)$ or belong to the $k$-th subspace can be written as

$$\sum_{i \in \mathcal{C}_k}\left(\frac{D}{m_i}\xi(\mathbf{x}_i, \Phi_i, \mathbf{U}_k, \mathbf{o}_k) + 1/|N_i|\sum_{j \in \mathcal{N}_i}\frac{D}{m_i}\xi(\mathbf{x}_j, \Phi_j, \mathbf{U}_k, \mathbf{o}_k)\right)$$

$$= \sum_{i=1}^{N}\mathbb{I}(i \in \mathcal{C}_k)\frac{D}{m_i}\xi(\mathbf{x}_i, \Phi_i, \mathbf{U}_k, \mathbf{o}_k) + 1/|N_i|\sum_{i=1}^{N}\mathbb{I}(i \in \mathcal{C}_k)\sum_{j=1}^{N}\mathbb{I}(j \in \mathcal{N}_i)\frac{D}{m_i}\xi(\mathbf{x}_j, \Phi_j, \mathbf{U}_k, \mathbf{o}_k)$$

$$= \sum_{i=1}^{N}\mathbb{I}(i \in \mathcal{C}_k)\frac{D}{m_i}\xi(\mathbf{x}_i, \Phi_i, \mathbf{U}_k, \mathbf{o}_k) + \sum_{j=1}^{N}1/|N_i|\sum_{i=1}^{N}\mathbb{I}(i \in \mathcal{C}_k)\mathbb{I}(j \in \mathcal{N}_i)\frac{D}{m_i}\xi(\mathbf{x}_j, \Phi_j, \mathbf{U}_k, \mathbf{o}_k)$$

$$= \sum_{i=1}^{N}\mathbb{I}(i \in \mathcal{C}_k)\frac{D}{m_i}\xi(\mathbf{x}_i, \Phi_i, \mathbf{U}_k, \mathbf{o}_k) + \sum_{j=1}^{N}n_k(j)\frac{D}{m_i}\xi(\mathbf{x}_j, \Phi_j, \mathbf{U}_k, \mathbf{o}_k)$$

$$= \sum_{i=1}^{N}\frac{D}{m_i}\xi(\mathbf{x}_i, \Phi_i, \mathbf{U}_k, \mathbf{o}_k)\left(\mathbb{I}(i \in \mathcal{C}_k) + n_k(i)\right)$$

where $n_k(i) = \sum_{j \in \mathcal{C}_k}1/|N_j|\mathbb{I}(i \in \mathcal{N}_j)$.

## A.3 Proof of Proposition 3

Since $\mathbf{o}_k$ is perpendicular to the subspace basis $\mathbf{U}_k$, we have $\mathbf{U}_k^{\mathrm{T}}\mathbf{o}_k = \mathbf{0}$. Since $\mathbf{o}_k'$ is located on the subspace, we have $\varepsilon(\mathbf{o}_k', \mathbf{U}_k, \mathbf{o}_k) = 0$. Thus,

$$\mathbf{o}_k' - \mathbf{U}_k(\mathbf{U}_k^{\mathrm{T}}\mathbf{U}_k)^{-1}\mathbf{U}_k^{\mathrm{T}}(\mathbf{o}_k' - \mathbf{o}_k) - \mathbf{o}_k = \mathbf{0}$$
$$\mathbf{o}_k' - \mathbf{U}_k(\mathbf{U}_k^{\mathrm{T}}\mathbf{U}_k)^{-1}\mathbf{U}_k^{\mathrm{T}}\mathbf{o}_k' - \mathbf{o}_k = \mathbf{0}$$
$$(\mathbf{I}_D - \mathbf{U}_k(\mathbf{U}_k^{\mathrm{T}}\mathbf{U}_k)^{-1}\mathbf{U}_k^{\mathrm{T}})\mathbf{o}_k' = \mathbf{o}_k$$

## A.4 Proof of Theorem 4

We first restate a corollary from Balzano et al. (2012). For the vector $\mathbf{x}$ and the subspace $\mathcal{S}^i$, $i \in \{0, 1, 2, ...\}$.

**Corollary 7.1.** *Let* $m > \frac{8}{3}\max_{i \neq 0}\left(d_i\rho(\mathcal{S}^i)\log{(\frac{2d_i}{\delta})}\right)$ *for fixed* $\delta > 0$. *Assume that* $\sin^2(\theta_0) < C_i(m)\sin^2(\theta_i), \forall i \neq 0$. *Then with probability at least* $1 - 4(k-1)\delta$, $\|\phi_\Phi - P_{S_\Omega^0}\phi_\Phi\|_2^2 < \|\phi_\Phi - P_{S_\Omega^i}\phi_\Phi\|_2^2, \forall i \neq 0$, *where* $\theta_0 = \sin^{-1}\left(\frac{\|\mathbf{x}-P_{S^0}\mathbf{x}\|_2}{\|\mathbf{x}\|_2}\right)$, $\theta_i = \sin^{-1}\left(\frac{\|\mathbf{x}-P_{S^i}\mathbf{x}\|_2}{\|\mathbf{x}\|_2}\right)$, $P_{S_\Omega^i} = \mathbf{U}_i(\mathbf{U}_i^T\mathbf{U}_i)^{-1}\mathbf{U}_i^T$, *and* $C_i(m) = \frac{m(1-\alpha_i)-d_i\mu(S^i)\frac{(1+\eta_i)^2}{1-\gamma_i}}{m(1+\alpha_0)}$.

Note that $C_k(m) \to 1$ as $m \to \infty$. The inequality $\sin^2(\theta_0) < C_i(m)\sin^2(\theta_i)$ illustrates the relation between vector $\mathbf{x}$, its corresponding subspace $\mathcal{S}^0$, and the subspace $\mathcal{S}^i$, $i \in \{1, 2, ...\}$. In this paper, we have the following conclusion about the frame $\mathbf{x}$, its corresponding subspace $\{\mathbf{U}_l, \mathbf{o}_l\}$, and other subspace $\{\mathbf{U}_k, \mathbf{o}_k\}$, $k \neq l, k \in \{1, 2, ..., K\}$, that is $\varepsilon(\mathbf{x}, \mathbf{U}_l, \mathbf{o}_l) < C_k\varepsilon(\mathbf{x}, \mathbf{U}_k, \mathbf{o}_k)$, where

$C_k = \frac{m(1-\alpha_k)-d_k\rho(\mathcal{S}^k)\frac{(1+\eta_k)^2}{1-\gamma_k}}{m(1+\alpha_0)} \frac{\|\mathbf{x}-\mathbf{o}_l\|_2^2}{\|\mathbf{x}-\mathbf{o}_k\|_2^2}$ and $\alpha_k, \eta_k, \gamma_k$ is the same as the variable in Corollary 7.1, which is defined in Theorem 1.

Similarly, the conclusion $\|\phi_\Phi - P_{S_\Omega^0}\phi_\Phi\|_2^2 < \|\phi_\Phi - P_{S_\Omega^i}\phi_\Phi\|_2^2$ in Corollary 7.1 is equal to the following form in this paper, i.e., $\varepsilon(\Phi\mathbf{x}, \Phi\mathbf{U}_l, \Phi\mathbf{o}_l) < \varepsilon(\Phi\mathbf{x}, \Phi\mathbf{U}_k, \Phi\mathbf{o}_k)$ for $\forall k \in \{1, 2, ..., K\}, k \neq l$. Then, we have the following conclusion $l = \underset{k\in\{1,2,...,K\}}{\arg\min} \varepsilon(\Phi\mathbf{x}, \Phi\mathbf{U}_k, \Phi\mathbf{o}_k)$ with $m > C_0$, where

$C_0 = \frac{8}{3}\max_k d_k\rho(\mathcal{S}^k)\log\left(\frac{2d_k}{\delta}\right).$

## B   PROOF OF LEMMA 5

We use the Non-commutative Bernstein Inequality as follows. Let $X_k = U_{\Phi(k)}U_{\Phi(k)}^{\mathrm{T}} - \frac{1}{r}I_r$, where the notation $U_{\Phi(k)}$ is as before, i.e. is the transpose of the $\Phi(k)^{th}$ th row of $U$, and $I_r$ is the $r \times r$ identity matrix. Note that this random variable is zero mean.

We must compute $\rho_k^2$ and $M$. Since $\Phi(k)$ is chosen uniformly with replacement, the $X_k$ are identically distributed, and $\rho$ does not depend on $k$. For ease of notation we will denote $U_{\Phi(k)}$ as $U_k$.

Using the fact that for positive semi-definite matrices, $\|A - B\|_2 \leq \max\{\|A\|_2, \|B\|_2\}$, and recalling again that $\|U_k\|_2^2 = \|U^{\mathrm{T}}e_k\|_2^2 = \|P_S e_k\|_2^2 \leq r\mu(\mathcal{S})/n$, we have

$$\|U_{\Phi(k)}U_{\Phi(k)}^{\mathrm{T}} - \frac{1}{r}I_r\|_2 \leq \max\{\frac{r\mu(\mathcal{S})}{D}, \frac{1}{D}\}$$

and we let $M := r\mu(\mathcal{S})/D$.

For $\rho$, we note

$$\|\mathbb{E}[X_kX_k^{\mathrm{T}}]\|_2 \leq \max\{\frac{r\mu(\mathcal{S})}{D^2}\|I_r\|_2, \frac{1}{D^2}\} = \frac{r\mu(\mathcal{S})}{D^2}$$

Thus we let $\rho := r\mu(\mathcal{S})/D^2$.

Now we can apply the Non-commutative Bernstein Inequality, Theorem 9. First we restrict $\tau$ to be such that $M\tau \leq m\rho^2$ to simplify the denominator of the exponent. Then we get that $2r\exp\left(\frac{-\tau^2/2}{2\rho^2+M\tau/3}\right) \leq 2r\exp\left(\frac{-\tau^2/2}{\frac{4}{3}m\frac{r\mu(\mathcal{S})}{D^2}}\right)$ and thus

$$\mathbb{P}\left[\left\|\sum_{k\in\Phi}(U_kU_k^{\mathrm{T}} - \frac{1}{D}I_r)\right\| > \tau\right] \leq 2r\exp\left(\frac{-3D^2\tau^2}{8mr\mu(\mathcal{S})}\right)$$

Now take $\tau = \gamma m/D$ with $\gamma$ defined in the statement of Theorem 1. Since $\gamma < 1$ by assumption, $M\tau \leq m\rho^2$ holds and we have

$$\mathbb{P}\left[\left\|\sum_{k\in\Phi}(U_kU_k^{\mathrm{T}} - \frac{1}{D}I_r)\right\| \leq \gamma m/D\right] \geq 1 - \delta$$

where $\delta = 2r\exp\left(\frac{-3D^2\tau^2}{8mr\mu(\mathcal{S})}\right)$. We note that $\left\|\sum_{k\in\Phi}U_kU_k^{\mathrm{T}} - \frac{m}{D}I_r\right\| \leq \gamma m/n$ implies that the minimum singular value of $\sum_{k\in\Phi}U_kU_k^{\mathrm{T}}$ is at least $(1-\gamma)m/D$. This in turn implies that $\left\|\left(\sum_{k\in\Phi}U_kU_k^{\mathrm{T}}\right)^{-1}\right\|_2 \leq \frac{D}{(1-\gamma)m}$, which completes the proof.

## C   PROOF OF LEMMA 6

We use McDiarmid's inequality in a very similar fashion to the proof of Lemma 1. Let $X_i = y_{\Phi(i)}U_{\Phi(i)}$, where $\Phi(i)$ refers to the $i$-th data index. Thus $y_{\Phi(i)}$ is a scalar, and the notation $U_{\Phi(i)}$ refers to an $r \times 1$ vector representing the transpose of the $\Phi(i)$ th row of $U$.

Let our function $f(X_1, X_2, ..., X_m) = \|\sum_{i=1}^m X_i\|_2 = \|U_\Phi^{\mathrm{T}} y_\Phi\|_2$. To find the $t_i$ of the theorem we first need to bound $\|X_i\|$ for all $i$. Observe that $\|U_{\Phi(i)}\|_2 = \|U^{\mathrm{T}} e_i\|_2 = \|P_S e_i\|_2 \le \sqrt{r\mu(\mathcal{S})/D}$ by assumption. Thus,

$$\|X_i\|_2 \le |y_{\Phi(i)}|\|U_{\Phi(i)}\|_2 \le \|y\|_\infty \sqrt{r\mu(\mathcal{S})/n}$$

Then observe $|f(X_1, X_2, ..., X_m) - f(X_1, X_2, ..., \hat{X}_k, ..., X_m)|$ is $\left|\left\|\sum_{i=1}^m X_i\right\|_2 - \left\|\sum_{i\ne k} X_i + \hat{X}_k\right\|_2\right| \le 2\|y\|_\infty \sqrt{\frac{r\mu(\mathcal{S})}{D}}$. Here, the first two inequalities follow from the triangle inequality. Next we calculate a bound for $\mathbb{E}[f(X_1, X_2, ..., X_m)] = \mathbb{E}[\|\sum_{i=1}^m X_i\|]$. Assume again that the points are taken uniformly with replacement. We have $\sum_{k=1}^r U_{jk}^2 = \|P_S e_j\|^2 \le \frac{r}{D}\mu(\mathcal{S})$, from which we can see that $\mathbb{E}\left[\left\|\sum_{i=1}^m X_i\right\|_2^2\right] \le \frac{m}{D}\frac{r\mu(\mathcal{S})}{D}\|y\|_2^2$.

Since $\mathbb{E}[\|X\|_2] \le \mathbb{E}[\|X\|_2^2]^{1/2}$ by Jensen's inequality, we have that $\mathbb{E}[\|\sum_{i=1}^m X_i\|_2] \le \sqrt{\frac{m}{D}}\sqrt{\frac{r\mu(\mathcal{S})}{D}}\|y\|_2$. Letting $\epsilon = \eta\sqrt{\frac{m}{D}}\sqrt{\frac{r\mu(\mathcal{S})}{D}}$ and plugging into Equation (15), we then have that the probability is bounded by $\exp\left(\frac{-2\eta^2 \frac{m}{D}\frac{r\mu(\mathcal{S})}{D}\|y\|_2^2}{4m\|y\|_\infty^2 \frac{r\mu(\mathcal{S})}{D}}\right)$. Thus, the resulting probability bound is

$$\mathbb{P}\left[\|U_\Phi y_\Phi\|_2^2 \ge (1+\eta)^2 \frac{mr\mu(\mathcal{S})}{D^2}\|y\|_2^2\right] \le \exp\left(\frac{-\eta^2 m\|y\|_2^2}{2D\|y\|_\infty^2}\right)$$

Substituting our definitions of $\mu(y)$ and $\eta$ shows that the lower bound holds with probability at least $1 - \delta$, where $\delta = \exp\left(\frac{-\eta^2 m\|y\|_2^2}{2D\|y\|_\infty^2}\right)$, completing the proof.

## D    PROOF OF LEMMA 7

**Theorem 8.** *(McDiarmid's Inequality McDiarmid et al. (1989)). Let $X_1, X_2, ..., X_n$ be independent random variables, and assume $f$ is a function for which there exist $t_i$, $i = 1, ..., n$ satisfying $\sup\limits_{x_1, x_2, ..., x_n, \hat{x}_i} |f(x_1, x_2, ..., x_n) - f(x_1, x_2, ..., \hat{x}_i, ..., x_n)| \le t_i$ where $\hat{x}_i$ indicates replacing the point value $x_i$ with any other of its possible values. Call $f(X_1, ..., X_n) := Y$. Then for any $\epsilon > 0$,*

$$\mathbb{P}[Y \ge \mathbb{E}[Y] + \epsilon] \le \exp\left(\frac{-2\epsilon^2}{\sum_{i==1}^n t_i^2}\right) \tag{15}$$

$$\mathbb{P}[Y \le \mathbb{E}[Y] - \epsilon] \le \exp\left(\frac{-2\epsilon^2}{\sum_{i=1}^n t_i^2}\right) \tag{16}$$

**Theorem 9.** *(Non-commutative Bernstein Inequality Gross et al. (2010); Recht (2011) ). Let $X_1, X_2, ..., X_m$ be independent zero-mean square $r \times r$ random matrices. Suppose $\rho_k^2 = \max\{\|\mathbb{E}[X_k^T X_k]\|_2, \|\mathbb{E}[X_k^T X_k]\|_2\}$ and $\|X_k\|_2 \le M$ almost surely for all $k$. Then for any $\tau > 0$,*

$$\mathbb{P}\left[\left\|\sum_{k=1}^m X_k\right\|_2 > \tau\right] \le 2r\exp\left(\frac{-\tau^2/2}{\sum_{k==1}^m \rho_k^2 + M\tau/3}\right)$$

To prove this we use McDiarmid's P inequality from Theorem 8 for the function $f(X_1, ..., X_m) = \sum_{i=1}^m X_i$. The resulting inequality is more commonly referred to as Hoeffding's inequality.

We begin with the first inequality. Set $X_i = y_{\Phi(i)}^2$. We seek a good value for $t_i$. Since $y_{\Phi(i)}^2 \le \|y\|_\infty^2$ for all $i$, we have

$$\left|\sum_{i=1}^m X_i - \sum_{i\ne k} X_i - \hat{X}_k\right| = |X_k - \hat{X}_k| \le 2\|y\|_\infty^2$$

We calculate $\mathbb{E}[\sum_{i=1}^{m} X_i]$ as follows. Define $\mathbb{I}_{\{\}}$ to be the indicator function, and assume that the points are taken uniformly with replacement.

$$\mathbb{E}[\sum_{i=1}^{m} X_i] = \mathbb{E}[\sum_{i=1}^{m} y_{\Phi(i)}^2] = \sum_{i=1}^{m} \mathbb{E}[\sum_{j=1}^{D} y_j^2 \mathbb{I}_{\{\Phi(i)=j\}}] = \frac{m}{D} \|y\|_2^2$$

Plugging into Equation (16), the left hand side is

$$\mathbb{P}[\sum_{i=1}^{m} X_i \leq \mathbb{E}[\sum_{i=1}^{m} X_i] - \epsilon] = \mathbb{P}[\sum_{i=1}^{m} X_i \leq \frac{m}{D} \|y\|_2^2 - \epsilon]$$

and letting $\epsilon = \alpha \frac{m}{D} \|y\|_2^2$, we then have that this probability is bounded by $\exp(\frac{-2\alpha^2 (\frac{m}{D})^2 \|y\|_2^4}{4m \|y\|_\infty^4})$ Thus, the resulting probability bound is

$$\mathbb{P}\left[ \|y_\Phi\|_2^2 \geq (1-\alpha) \frac{m}{D} \|y\|_2^2 \right] \geq 1 - \exp\left( \frac{-\alpha^2 m \|y\|_2^4}{2D^2 \|y\|_\infty^4} \right)$$

Substituting our definitions of $\mu(y)$ and $\alpha$ shows that the lower bound holds with probability at least $1 - \delta$, where $\delta = \exp\left( \frac{-\alpha^2 m \|y\|_2^4}{2D^2 \|y\|_\infty^4} \right)$. The argument for the upper bound is identical after replacing Equation (15) instead of (16). The Lemma now follows by applying the union bound