# OpenReview forum: "Exploring Temporal Semantic for Incomplete Clustering"
_ICLR.cc/2025/Conference — Submitted to ICLR 2025_

### Official Review · Reviewer_ivFX · 2024-11-01

**Soundness:** 3
**Presentation:** 4
**Contribution:** 3
**Rating:** 6
**Confidence:** 3

**Summary:**

Comments:
The manuscript presents a novel clustering framework, ETC-IC, intended to tackle the issue of clustering sequential data with incomplete features. The topic is of significant contemporary relevance, given the increasing focus on data with missing attributes in the clustering domain.

Weaknesses:
1.	It is recommended to provide a summary of the entire work through a framework figure.
2.	The model is evaluated solely on the human motion dataset, and thus this work is validated within the human motion domain. Consequently, the current title is not appropriate, authors should revise it to ‘human motion learning’ or evaluate this model in a broader context.
3.	In the ABLATION STUDY section, the authors should provide a detailed explanation of the ablation experiment setup and non-temporal semantics to validate the module's effectiveness.

**Strengths:**

This paper is well written.

**Weaknesses:**

Weaknesses:
1.	It is recommended to provide a summary of the entire work through a framework figure.
2.	The model is evaluated solely on the human motion dataset, and thus this work is validated within the human motion domain. Consequently, the current title is not appropriate, authors should revise it to ‘human motion learning’ or evaluate this model in a broader context.
3.	In the ABLATION STUDY section, the authors should provide a detailed explanation of the ablation experiment setup and non-temporal semantics to validate the module's effectiveness.

**Questions:**

1.	It is recommended to provide a summary of the entire work through a framework figure.
2.	The model is evaluated solely on the human motion dataset, and thus this work is validated within the human motion domain. Consequently, the current title is not appropriate, authors should revise it to ‘human motion learning’ or evaluate this model in a broader context.
3.	In the ABLATION STUDY section, the authors should provide a detailed explanation of the ablation experiment setup and non-temporal semantics to validate the module's effectiveness.

---

### Official Review · Reviewer_KEMZ · 2024-11-02

**Soundness:** 3
**Presentation:** 3
**Contribution:** 3
**Rating:** 6
**Confidence:** 3

**Summary:**

This paper introduces ETC-IC framework, which possesses the capability to seamlessly integrate temporal information while concurrently addressing the challenge of missing data. Firstly, to manage the issue of missing entries, ETC-IC employs an algebraic subspace analysis and develop a theoretically grounded alternative, thereby ensuring accurate clustering even in the presence of  incomplete data. Secondly, ETC-IC explores the temporal semantics inherent in sequential data by aligning data points and their temporal assignments through a temporal semantic consistency constraint, thereby ensuring that data points with similar temporal semantics are clustered together. The handling of missing data and the exploration of temporal semantics are unified within a single cohesive framework, thereby demonstrating the adaptability and versatility of the proposed method in addressing incomplete sequential data as required.

**Strengths:**

1. This paper presents a clustering framework distinguished by its remarkable adaptability in addressing the inherent challenges posed by incomplete sequential data.
2. We introduce an innovative temporal semantic consistency constraint, which markedly enhances the efficacy of subspace clustering for sequential data.
3. We provide a rigorous theoretical analysis, enabling an equivalent approach even in the presence of missing data, whilst effectively exploring temporal semantics.

**Weaknesses:**

1. In this paper, while the algorithm has already been exhaustively described and experimentally validated, it is recommended to include an analysis of the algorithm's time complexity to further enhance the completeness.
2. it is recommended to incorporate additional evaluation metrics to further strengthen the assessment of its performance.

**Questions:**

1. Is the objective function convex, and if so, suggest adding a proof of convergence analysis instead of just giving a figure 6?
2. Did the authors use five datasets or four? Why is it four at one time and five at another?

---

### Official Review · Reviewer_42Da · 2024-11-02

**Soundness:** 2
**Presentation:** 2
**Contribution:** 2
**Rating:** 3
**Confidence:** 3

**Summary:**

This paper introduced a novel clustering framework called Exploring Temporal Semantic for Incomplete Clustering (ETC-IC), which leverages temporal information within sequential data to enhance the clustering accuracy. Unlike previous works, ETC-IC clusters data without requiring prior imputation, making the results less sensitive to missing data attribute issues. This work valid ETC-IC on 5 human motion benchmarks and the proposed model consistently surpasses current SOTA methods.

**Strengths:**

(S1) The work proposes a novel angle to cluster human motion data considering data temporal sequence.
(S2) It outperforms the previous baselines when adapted with the ‘missing entries’ data processing proposed by this work.

**Weaknesses:**

(W1) Experiment -- The experiments are not sufficiently well designed. Firstly, they adapt the proposed MAR and MNAR directly to the previous baselines without comparing the baselines’ original performances. Secondly, one of the core concept in this paper: missing entries, are manually designed and generated rather than an off-the-shelf nature existing in the dataset. The results should also show how the model behaves on samples without missing entries to show the model’s general capacity. Thirdly, the quantitative analysis is poor, where Figure 5 only showcases some samples while Figure 7 is a case study rather than quantitative analysis.

(W2) Presentation -- Moreover, the presentation of the experiments is with many errors. Firstly, both Figure 1 and Figure 8 report results on 5 datasets, but the figure caption and paper introduce there are only 4 datasets instead of 5. Secondly, the introduction to Mocap is poor, where it doesn’t introduce clearly what are the sequence data. Thirdly, Fig 2 (b) and Fig 3 (b) are with low quality. There is a clear obstruction between plot lines and bottom-left frames. Fourthly, the ablation study is reported in a Table but presented as a Figure (Figure 8). Also, the caption is on top of figure 8 while all the other captions for Figures stay in the bottom.

(W3) Lack of guidance -- Besides the experiments, it’s also very hard to comprehend the authors formal derivation, as there are very few intuitive explanations. Without a clear guidance and introduction to the data, the author directly introduces the formulas without elaborating how the symbols are connected to the data in this work. Later on, in the experiment, only a simple ablation study in Figure 8 shows the effectiveness of ‘temporal semantics’, while it doesn’t analyze the effectiveness of theorem1 to 4 respectively, leaving it unknown which part of the theorem truly works, and which part might fail. Also, the theorem 3 is missing, where it is replaced by ‘proposition 3’.

**Questions:**

(Q1) Why are the Figure 3 to Figure 6 only reported on a single dataset while ignoring the other 4 selected datasets?

(Q2) Please introduce the data in a more structured way before section 3. A clearer definition should also be introduced to show what do you mean by ‘temporal semantics’ and why they are important, instead of talking about concept in a high-level way.

(Q3) Please also include the description of temporal semantic in algorithm 1 to explicitly make people know how it works.

---

### Official Review · Reviewer_m58G · 2024-11-03

**Soundness:** 2
**Presentation:** 2
**Contribution:** 2
**Rating:** 3
**Confidence:** 3

**Summary:**

The paper presents a method for clustering of sequential data with missing values. It incorporates temporal constraints to model the expected continuity across time. It characterises the method theoretically, and evaluates on different motion segmentation benchmark datasets.

**Strengths:**

S1. The paper addresses an important problem of handling missing values.
S2. The empirical evaluation makes use of five different benchmarking data sets for motion segmentation.

**Weaknesses:**

W1. The presentation of the paper is weak, lacking discussion, which makes it hard to follow. Concretely, the scope of the paper is unclear initially, going from clustering of missing features to subspace clustering and back. After that, the method and its properties are presented, without discussing how it relates to existing work, what motivates each step, and whether there are design alternatives.
W2. The discussion of related work fails to convey which methods are related in terms of applicability to the problem under study or in terms of method similarities, and which are less closely related. Instead, the related work section lists technical aspects of different methods, without any assessment as to their suitability for the problem under study.
W3. The empirical study is weak. In the experimental evaluation, only methods for clustering with missing data are studied, but none for clustering temporal data. As the datasets under study are characterised by strong temporal signals, the competitors are thus very weak baselines.
W4. The paper fails to provide sufficient information about the setup in the experiments, and some details about the method are confusing. For example, for the experiment in Fig. 2, linear interpolation is used prior to running the method. This seems to contradict the purpose of the method of being able to handle missing data. Also, it is unclear if linear interpolation was also used prior to running the competitors. This should be clarified in the description, and experiments comparing with and without interpolation should be conducted.
W5. The accuracy in several of the experiments is very high, close to 80%, even when half of the data is missing - indicating that the problem might be too easy for any temporal method (as stated above, none of them are considered here). The experimental evaluation should thus include temporal clustering methods as well, possibly using interpolation if necessary (as in W4). More challenging datasets, where missing data has a stronger impact on accuracy should be studied in order to understand the robustness of the method.
W6. It is unclear how quickly the method converges in general, only one example is provided for one of the datasets. The paper should provide convergence results across datasets and runs.

**Questions:**

Why do you not include any temporal clustering methods (possibly after interpolation)? How do temporal clustering methods perform on this type of data?
Which of the methods in related work are applicable to your evaluation scenario? You could consider including a table that lists core properties, indicating which of them are met by which competitor, instead of (long) textual descriptions.

---

### Meta-Review · Area_Chair_j4mr · 2024-12-16

**Metareview:**

This paper focuses on clustering sequential data with missing attributes. Unlike conventional heuristic methods, this paper leverages temporal information within sequential data, and directly clusters incomplete data under rigorous theoretical guarantees. It employs an algebraic subspace analysis and develops a theoretically grounded alternative.  The handling of missing data and the exploration of temporal semantics are unified into a shared framework. Experimental results illustrate the effectiveness.

The experiments are not sufficiently well designed. The original performance of baselines is not compared, and the quantitative analysis is limited. The presentation of experiments is with many errors. Moreover, there is a lack of any temporal clustering methods. The authors directly introduce the formulas without elaborating how the symbols are connected to the data in this work, which leads to it very hard to understand. Besides, the analysis of the algorithm's time complexity is missing.  The ablation experiment setup is not detailed. Also, there is no any rebuttal.

**Additional Comments On Reviewer Discussion:**

The experiments are not sufficiently well designed. The original performance of baselines is not compared, and the quantitative analysis is limited. The presentation of experiments is with many errors. Moreover, there is a lack of any temporal clustering methods. The authors directly introduce the formulas without elaborating how the symbols are connected to the data in this work, which leads to it very hard to understand. Besides, the analysis of the algorithm's time complexity is missing.  The ablation experiment setup is also not detailed.

---

### Decision · Program_Chairs · 2025-01-22

Reject